# VO$_{2max}$ prediction based on submaximal cardiorespiratory relationships and body composition in male runners and cyclists: a population study

Szczepan Wiecha[1]*, Przemysław Seweryn Kasiak[2], Piotr Szwed[3], Tomasz Kowalski[4], Igor Cieśliński[1], Marek Postuła[3], Andrzej Klusiewicz[1]

[1]Department of Physical Education and Health in Biala Podlaska, Faculty in Biala Podlaska, Jozef Pilsudski University of Physical Education, Warsaw, Poland; [2]Students' Scientific Group of Lifestyle Medicine, 3rd Department of Internal Medicine and Cardiology, Medical University of Warsaw, Warsaw, Poland; [3]Department of Experimental and Clinical Pharmacology, Center for Preclinical Research and Technology CEPT, Medical University of Warsaw, Warsaw, Poland; [4]Institute of Sport-National Research Institute, Warsaw, Poland

## Abstract

**Background:** Oxygen uptake (VO$_2$) is one of the most important measures of fitness and critical vital sign. Cardiopulmonary exercise testing (CPET) is a valuable method of assessing fitness in sport and clinical settings. There is a lack of large studies on athletic populations to predict VO$_{2max}$ using somatic or submaximal CPET variables. Thus, this study aimed to: (1) derive prediction models for maximal VO$_2$ (VO$_{2max}$) based on submaximal exercise variables at anaerobic threshold (AT) or respiratory compensation point (RCP) or only somatic and (2) internally validate provided equations.

**Methods:** Four thousand four hundred twenty-four male endurance athletes (EA) underwent maximal symptom-limited CPET on a treadmill (n=3330) or cycle ergometer (n=1094). The cohort was randomly divided between: variables selection (n$_{runners}$ = 1998; n$_{cyclist}$ = 656), model building (n$_{runners}$ = 666; n$_{cyclist}$ = 219), and validation (n$_{runners}$ = 666; n$_{cyclist}$ = 219). Random forest was used to select the most significant variables. Models were derived and internally validated with multiple linear regression.

**Results:** Runners were 36.24±8.45 years; BMI = 23.94 ± 2.43 kg·m$^{-2}$; VO$_{2max}$=53.81±6.67 mL·min$^{-1}$·kg$^{-1}$. Cyclists were 37.33±9.13 years; BMI = 24.34 ± 2.63 kg·m$^{-2}$; VO$_{2max}$=51.74±7.99 mL·min$^{-1}$·kg$^{-1}$. VO$_2$ at AT and RCP were the most contributing variables to exercise equations. Body mass and body fat had the highest impact on the somatic equation. Model performance for VO$_{2max}$ based on variables at AT was $R^2$=0.81, at RCP was $R^2$=0.91, at AT and RCP was $R^2$=0.91 and for somatic-only was $R^2$=0.43.

**Conclusions:** Derived prediction models were highly accurate and fairly replicable. Formulae allow for precise estimation of VO$_{2max}$ based on submaximal exercise performance or somatic variables. Presented models are applicable for sport and clinical settling. They are a valuable supplementary method for fitness practitioners to adjust individualised training recommendations.

**Funding:** No external funding was received for this work.

*For correspondence:
szczepan.wiecha@awf.edu.pl

## Editor's evaluation

The authors have established new formulas to predict maximum oxygen uptake for cyclists and runners based on submaximal exercise testing and anthropometric characteristics. This is an important study with a large and comprehensive dataset, which may be helpful for many exercise labs. The work is convincing, using appropriate and validated methodology in line with the current state-of-the-art, as shown by references to common exercise books.

## Introduction

The oxygen uptake ($VO_2$) is considered an important metric in assessing cardiorespiratory fitness, health status, or endurance performance potential (*Guazzi et al., 2012*). With the application of standardised procedures and interpretation protocols, during graded exercise tests (GXT), the (maximal oxygen uptake) $VO_{2max}$ can be established (*Bentley et al., 2007*). GXT is the most widely used assessment to examine the dynamic relationship between exercise and integrated physiological systems (*Albouaini et al., 2007*; *Bentley et al., 2007*). The information from GXT during cardiopulmonary exercise testing (CPET) can be applied across the spectrum of sport performance, occupational safety screening, research, and clinical diagnostics (*Guazzi et al., 2017*).

$VO_{2\ max}$ is often used as a boundary between severe and extreme intensity domains and by definition requires maximal effort from the tested subject (*Gaesser and Poole, 1996*). However, it is not always recommended or possible to undertake a test to exhaustion (*Guazzi et al., 2012*). For the athletes, the proximity of competition or injury history can allow submaximal testing, but not testing to exhaustion (*Sassi et al., 2006*). Testing that requires maximal effort may be disruptive to the training process or interfere with race performance (*Coutts et al., 2007*; *Lamberts et al., 2011*). Due to practical constraints, tests to exhaustion or peak-power-output tests are often performed only two or three times a year (*Coutts et al., 2007*).

However, $VO_2$ values are widely used in sport science and the decision-making process (*Mann et al., 2013*). $VO_2$ is widely considered one of the major endurance performance determinants (*Joyner and Coyle, 2008*). Using $VO_{2max}$ to guide the selection process, prescribing training intensity, assessing training adaptations, or predicting race times is a common practice in high-performance sports (*Bassett and Howley, 2000*; *Bentley et al., 2007*; *Hawley and Noakes, 1992*; *Noakes et al., 1990*).

$VO_{2max}$ is also one of the critical vital signs coordinating the function of the cardiovascular, respiratory, and muscular systems, it is an indicator of overall body health status (*Kaminsky et al., 2017*). Quantifying $VO_{2max}$ provides additional input regarding clinical decision-making, risk stratification, evaluation of therapy, and physical activity guidelines (*Guazzi et al., 2012*). For patients undertaking a test to exhaustion is rarely needed or possible due to health restraints or cardiac risk (*Guazzi et al., 2016*).

For many years researchers have studied indirect methods of estimating $VO_{2max}$(*Sartor et al., 2013*). Protocols such as the Astrand-Ryhming Test, Six-Minute Walk Test, or YMCA Step Test have been established and validated (*Astrand and Ryhming, 1954*; *Beutner et al., 2015*; Carey, 2022; *Jalili et al., 2018*). Moreover, estimation of the $VO_2$ and heart rate (HR) values below the ventilatory threshold can be based on cardiorespiratory kinetics assessment using randomised changes in the work rate known as a pseudo-random binary sequences testing (*Hoffmann et al., 2022*). However, with the development of technology, the accessibility of laboratory testing and mobile testing improved (*Montoye et al., 2020*; *Pritchard et al., 2021*). Therefore, new opportunities to develop more precise yet simple and accessible methods and models to assess $VO_{2max}$ occur (*Jurov et al., 2023*). This appears to be especially important considering the low prediction accuracy of most of the $VO_{2max}$ formulae that were validated in our previous study (*Wiecha et al., 2023*).

Recently, we have been observing the development of prediction methods with the usage of machine learning (ML) and artificial intelligence (AI) (*Ashfaq et al., 2022*). Both ML and AI are used in sport science as forecasting and decision-making support tools (*Abut and Akay, 2015*; *Bobowik and Wiszomirska, 2022*; *Chmait and Westerbeek, 2021*; *Hammes et al., 2022*; *Rossi et al., 2021*). There is growing evidence that $VO_{2max}$ prediction based on ML models, especially support vector ML and artificial neural network models, exhibits more robust and accurate results compared to MLR only (*Abut and Akay, 2015*; *Ashfaq et al., 2022*).

Therefore, in this research, with the support of ML, we look for algorithms and prediction patterns that allow us to use values obtained during submaximal CPET and somatic measurements to estimate maximal $VO_{2max}$ values in male runners and cyclists. We stipulate that prediction models allow for accurate calculation of $VO_{2max}$ based on somatic or submaximal CPET variables.

## Materials and methods

We have applied the development and validation of the prediction TRIPOD guidelines to conduct the study (see TRIPOD Checklist for Prediction Model Development and Validation) (*Collins et al., 2015*). The study is based on retrospective data analysis from the CPET registry collected from 2013 to 2021 at the medical clinic (Sportslab, Warsaw, Poland). All CPET have been performed at the individual request of participants, as a part of regular training monitoring or performance assessment.

### Ethical approval

The Institutional Review Board of the Bioethical Committee at the Medical University of Warsaw (AKBE/32/2021) has approved the study protocol. The regulations of the Declaration of Helsinki were met during all parts of the study. Each study participant delivered written consent to undergo CPET and participate in the study.

### Derivation cohort

We selected the cohort with the use of rigorous exclusion/inclusion criteria. Due to the insufficient number of women in our database and the number of potential variables in the regression models for adequate power, we had to limit ourselves to conduct analysis in the male population only (*Martens and Logan, 2021*). Out of 6439 healthy, adult male cyclists and long-distance runners that undergone CPET, 4423 met the criteria as further: (1) age ≥18 years, (2) declared regular cycling or running training for ≥3 months, (3) had no extreme outliers ≤ or ≥±3 standard deviations (SD) from mean for all of the testing variables (beyond ≥±3 SD in $VO_{2max}$), (4) lack of any injury, medical condition, or addiction in medical history that may affect exercise capacity, (5) not taking any medications with a modifying effect on exercise capacity, (6) maximum exertion achieved during CPET. We defined the maximum exertion in CPET as the fulfilment of the minimum six of the following criteria: (1) respiratory exchange ratio (RER) ≥1.10, (2) present $VO_2$ plateau (growth <100 mL·min$^{-1}$ in $VO_2$ despite increased running speed or cycling power), (3) respiratory frequency (fR) ≥45 breaths·min$^{-1}$, (4) declared subjective exertion intensity during CPET ≥18 in the Borg scale (*Borg, 1970*), (5) blood lactate concentration $[La^-]_b$ ≥8 mmol·L$^{-1}$, (6) growth in speed/power ≥10% of respiratory compensation point (RCP) values after exceeding the RCP, (7) peak heart rate (HRpeak) ≥15 beats·min$^{-1}$ below predicted maximal heart rate (HR$_{max}$) (*Lach et al., 2021*).

Participants' selection procedure has been shown in *Figure 1*.

### Somatic measurements and CPET protocols

Body mass was measured with a body composition (BC) analyser (Tanita, MC 718, Japan) with the multifrequency of 5 kHz/50 kHz/250 kHz via the bioimpedance analysis and normal testing mode. The participants' skin was cleaned with alcohol before placing the electrodes on the skin. Prior to the test, the participants received instructions to refrain from exercising for 2 hr, consume a light meal rich in carbohydrates 2–3 hr beforehand, and maintain hydration by drinking isotonic beverages. Additionally, they were advised to abstain from medications, caffeine, and cigarettes on the day of the test.

Running CPET (TE) was performed on a mechanical treadmill (h/p/Cosmos Quasar, Germany). Cycling CPET (CE) was performed on Cyclus-2 (RBM elektronik-automation GmbH, Leipzig, Germany). Hans Rudolph V2 mask (Hans Rudolph, Inc, Shawnee, KS, USA), breath-by-breath method with Cosmed Quark CPET gas exchange analysing device (Cosmed Srl, Rome, Italy), and Quark PFT Suite to Omnia 1.6 software were utilised. The gas analyser device was regularly calibrated with the reference gas (16% $O_2$; 5% $CO_2$) in accordance with the manufacturer's instructions (Airgas USA, LLC, Plumsteadville, PA, USA). From 2013 to 2021, three Cosmed Quark CPET units were used. HR was measured with the Cosmed torso belt (Cosmed srl, Rome, Italy). $[La^-]_b$ was measured via enzymatic-amperometric electrochemical technique with Super GL2 analyser (Müller Gerätebau GmbH, Freital, Germany). The $[La^-]_b$ analyser was regularly calibrated before each measurement series. The 40 m$^2$ indoor, air-conditioned

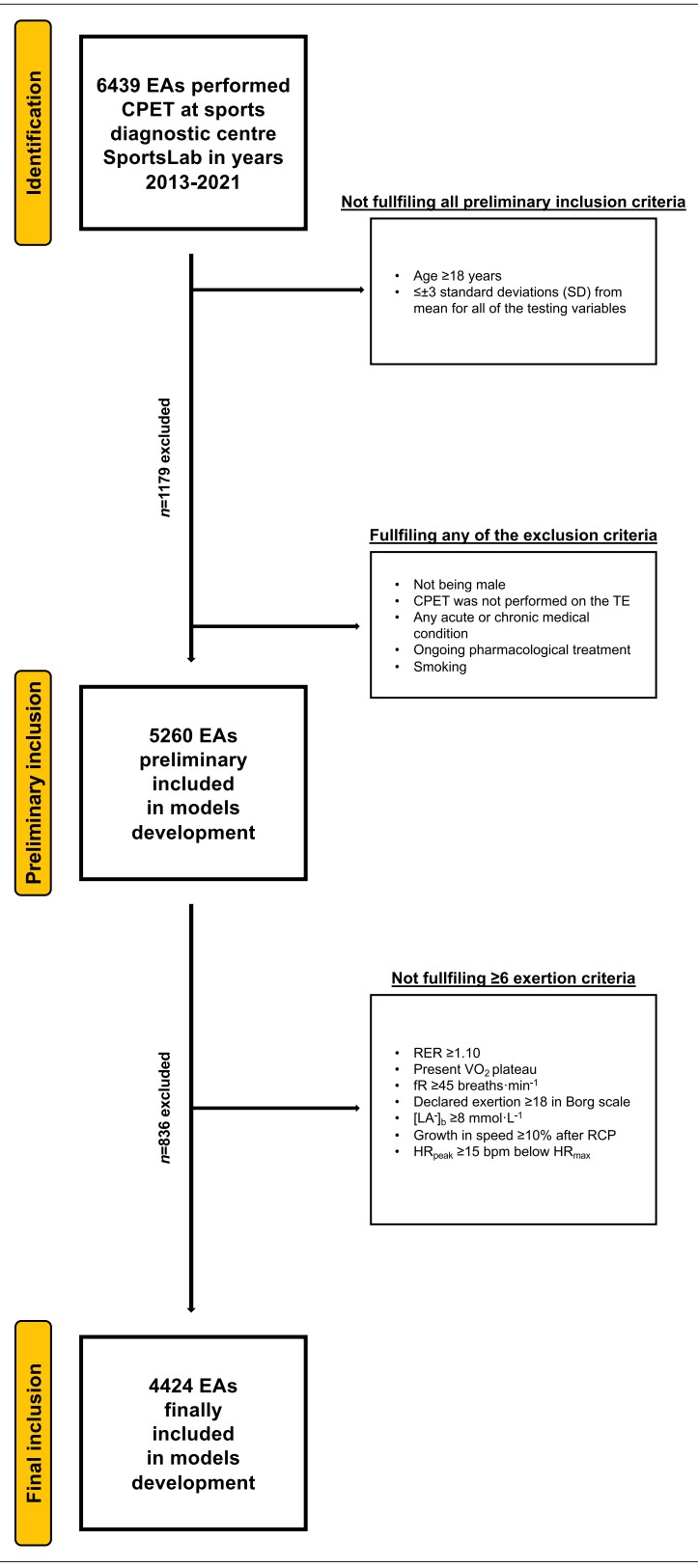

**Figure 1.** Flowchart of the preliminary inclusion and exclusion process. Abbreviations: EA, endurance athlete; CPET, cardiopulmonary exercise testing; SD, standard deviation; TE, treadmill; RER, respiratory exchange ratio; $VO_2$, oxygen uptake ($mL·min^{-1}·kg^{-1}$); $[La^-]_b$, lactate concentration ($mmol·L^{-1}$); fR, breathing frequency ($breaths·min^{-1}$); RCP, respiratory compensation point; $HR_{peak}$, peak heart rate ($beats·min^{-1}$); $HR_{max}$, maximal heart rate (bpm). At both stages of the selection, some participants met several (>1) exclusion criteria.

laboratory with 20–22°C temperature and 40–60% humidity, and 100 m ASL provided the same conditions for all BC and CPET.

Each CPET began with a 5 min personalised warm-up (walk or easy jog with 'conversational' intensity for running, easy pedalling with 'conversational' intensity for cycling). Then after the preparation (about 5 min), the continuous progressive step test was conducted. Due to the population diversity (training status), the running test speed started from 7 to 12 km·hr$^{-1}$ with a 1% treadmill incline. The choice of initial starting speed was determined by the interview and sports results achieved. For example, those running less than 60 min at a distance of 10 km started the test at 7 km/hr, while those running 10 km for less than 35 min started the test at an initial speed of 12 km/hr. The pace increased by 1 km·hr$^{-1}$ every 2 min with no change in incline. The cycling test began at 60–150 W, depending on the athletes training status. The power increased by 20–30 W every 2 min. It was recommended to maintain a constant cadence of 80–90 (repetition·min$^{-1}$) during the test. The tests were terminated due to exhaustion: volitional inability to continue the activity or/and VO$_2$ and HR plateau with increasing load or/and observed disturbance of coordination in running or/and inability to maintain the set cadence. Due to the graded protocol used, the cycling power and running speed values have been calculated as a function of time to better reflect the actual level for the test moment being determined (*Kuipers et al., 1985*). Before the test, after every step, and 3 min after the termination of the effort technician took a 20 µL blood sample from a fingertip. Samples were collected during the test without interrupting the effort. The samples were taken from the initial puncture. The first blood drop was collected into the swab and the second blood drop was drawn for further analysis into the capillary. VO$_{2max}$ was recorded as the highest value (15 s intervals) before the termination of the test. HR$_{max}$ was recorded as the highest value obtained at the end of the test, without averaging.

The anaerobic threshold (AT) was established with the following criteria: (1) common start of VE/VO$_2$ and VE/VCO$_2$ curves, (2) end-tidal partial pressure of oxygen raised constantly with the end-tidal partial pressure of carbon dioxide (*Beaver et al., 1986*). The was established with the following criteria: (1) PetCO$_2$ must decrease after reaching maximal amount, (2) the presence of fast nonlinear growth in VE (second deflection), (3) the VE/VCO$_2$ ratio achieved minimum and started to rise, and (4) a nonlinear increase in VCO$_2$ versus VO$_2$ (lack of linearity) (*Beaver et al., 1986*). The [La$^-$]$_b$ was estimated for AT and RCP in relation to power or speed (*Wiecha et al., 2022*).

## Data analysis

Our comprehensive ML approach enables the evaluation of each formula by preliminary variables precision (at the stage of selection), then accuracy (during the model's building) and recall (in internal validation).

Individual CPET results were saved into the Excel file (Microsoft Corporation, Redmond, WA, USA) and a custom-made script was used to generate the database in Excel (Python programming). Further, mean, SD, and 95% confidence intervals (CI) were calculated. The normality of the distribution of the

**Table 1.** Basic anthropometric characteristics for runners.

| Variable (unit) | Derivation group n=1998 | | | Testing group n=666 | | | Validation group n=666 | | |
|---|---|---|---|---|---|---|---|---|---|
| | Mean | CI | SD | Mean | CI | SD | Mean | CI | SD |
| Age (years) | 36.2 | 35.6–36.9 | 8.45 | 35.9 | 35.5–36.3 | 8.05 | 35.5 | 34.9–36.2 | 8.14 |
| Height (cm) | 180.0 | 179.6–180.5 | 6.04 | 179.4 | 179.1–179.7 | 6.13 | 179.7 | 179.2–180.2 | 6.61 |
| BM (kg) | 77.7 | 77.0–78.4 | 9.35 | 77.7 | 77.3–78.1 | 9.29 | 77.9 | 77.1–78.6 | 10.1 |
| BMI (kg·m$^{-2}$) | 23.9 | 23.8–24.1 | 2.43 | 24.1 | 24.0–24.2 | 2.41 | 24.1 | 23.9–24.3 | 2.56 |
| BF (%) | 15.4 | 15.1–15.7 | 4.55 | 15.5 | 15.3–15.7 | 4.52 | 15.4 | 15.1–15.8 | 4.55 |
| FM (kg) | 12.2 | 11.9–12.6 | 4.68 | 12.3 | 12.1–12.5 | 4.65 | 12.3 | 11.9–12.7 | 4.92 |
| FFM (kg) | 65.5 | 65.0–66.0 | 6.43 | 65.4 | 65.1–65.7 | 6.31 | 65.6 | 65.1–66.1 | 6.86 |

BM = body mass. BMI = body mass index. BF = body fat. FM = fat mass. FFM = fat-free mass. CI = 95% confidence interval. SD = standard deviation.

**Table 2.** Basic anthropometric characteristics for cyclists.

| Variable (unit) | Derivation group n=656 | | | Testing group n=219 | | | Validation group n=219 | | |
|---|---|---|---|---|---|---|---|---|---|
| | Mean | CI | SD | Mean | CI | SD | Mean | CI | SD |
| Age (years) | 37.3 | 36.6–38.0 | 9.13 | 37.1 | 35.9–38.4 | 9.50 | 37.6 | 36.5–38.8 | 8.46 |
| Height (cm) | 179.9 | 179.4–180.4 | 6.27 | 180.1 | 179.2–181.0 | 6.96 | 180.2 | 179.4–181.0 | 6.13 |
| BM (kg) | 78.8 | 78.1–79.6 | 9.80 | 79.1 | 77.7–80.5 | 10.4 | 79.8 | 78.4–81.3 | 10.9 |
| BMI (kg·m$^{-2}$) | 24.3 | 24.1–24.6 | 2.63 | 24.4 | 24.0–24.7 | 2.80 | 24.6 | 24.2–25.0 | 2.96 |
| BF (%) | 16.4 | 15.7–17.1 | 4.99 | 16.1 | 15.7–16.5 | 4.81 | 16.2 | 15.5–16.8 | 4.87 |
| FM (kg) | 13.3 | 12.6–14.1 | 5.66 | 13.0 | 12.6–13.4 | 5.27 | 13.3 | 12.5–14.0 | 5.85 |
| FFM (kg) | 65.8 | 64.9–66.6 | 6.25 | 65.8 | 65.4–66.3 | 6.06 | 66.6 | 65.7–67.4 | 6.58 |

BM = body mass. BMI = body mass index. BF = body fat. FM = fat mass. FFM = fat-free mass. CI = 95% confidence interval. SD = standard deviation.

data was examined using the Shapiro-Wilk test and intergroup differences were calculated using the Student's t-test for independent variables.

Three-step variable selection procedures based on random forests were applied using the R package VSURF in RStudio software (R Core Team, Vienna, Austria; version 3.6.4) (*Genuer et al., 2016*). For each level of measurement (AT, RCP) and their combination (AT+RCP), significant variables were identified separately. The first step was dedicated to eliminate irrelevant variables from the dataset. Second step aimed to select all variables related to the response for interpretation purposes. The third step refined the selection by eliminating redundancy in the set of variables selected by the second step, for prediction purposes (*Genuer et al., 2017*). Each time for variables selection,

**Table 3.** Cardiopulmonary exercise testing (CPET) characteristics for runners.

| Variable (unit) | Derivation group n=1998 | | | Testing group n=666 | | | Validation group n=666 | | |
|---|---|---|---|---|---|---|---|---|---|
| | Mean | CI | SD | Mean | CI | SD | Mean | CI | SD |
| rVO$_{2AT}$ (mL·min$^{-1}$·kg$^{-1}$) | 38.4 | 38.1–38.8 | 5.01 | 38.5 | 38.3–38.7 | 4.88 | 38.1 | 37.7–38.5 | 5.16 |
| RER$_{AT}$ | 0.87 | 0.86–0.87 | 0.04 | 0.87 | 0.86–0.87 | 0.04 | 0.87 | 0.86–0.87 | 0.04 |
| HR$_{AT}$ (beats·min$^{-1}$) | 151.5 | 150.8–152.3 | 10.3 | 151.0 | 150.6–151.5 | 10.8 | 152.0 | 151.2–152.8 | 10.8 |
| VE$_{AT}$ (L·min$^{-1}$) | 79.1 | 78.1–80.0 | 12.2 | 78.3 | 77.8–78.9 | 12.0 | 77.2 | 76.3–78.2 | 12.0 |
| SPEED$_{AT}$ (km·h$^{-1}$) | 11.0 | 10.9–11.1 | 1.45 | 11.0 | 11.0–11.1 | 1.36 | 10.9 | 10.8–11.0 | 1.42 |
| LA$_{AT}$ (mmol·L$^{-1}$) | 2.08 | 2.02–2.14 | 0.63 | 1.80 | 1.76–1.83 | 0.62 | 2.35 | 2.27–2.42 | 0.72 |
| rVO$_{2RCP}$ (mL·min$^{-1}$·kg$^{-1}$) | 47.5 | 47.0–48.0 | 5.88 | 47.7 | 47.4–48.0 | 6.15 | 47.3 | 46.8–47.8 | 6.16 |
| RER$_{RCP}$ | 1.00 | 1.00–1.00 | 0.04 | 1.00 | 1.00–1.00 | 0.04 | 1.00 | 1.00–1.00 | 0.03 |
| HR$_{RCP}$ (beats·min$^{-1}$) | 173.4 | 172.7–174.1 | 9.21 | 173.2 | 172.8–173.6 | 9.30 | 174.3 | 173.5–175.0 | 9.50 |
| VE$_{RCP}$ (L·min$^{-1}$) | 114.7 | 113.5–116.0 | 15.9 | 113.9 | 113.1–114.6 | 16.7 | 112.7 | 111.4–114.0 | 16.2 |
| SPEED$_{RCP}$ (km·h$^{-1}$) | 14.0 | 13.9–14.1 | 1.77 | 14.1 | 14.0–14.1 | 1.70 | 13.9 | 13.8–14.1 | 1.75 |
| LA$_{RCP}$ (mmol·L$^{-1}$) | 4.72 | 4.63–4.82 | 1.04 | 4.40 | 4.34–4.45 | 1.04 | 4.81 | 4.69–4.93 | 1.14 |
| rVO$_{2max}$ (mL·min$^{-1}$·kg$^{-1}$) | 53.8 | 53.3–54.3 | 6.67 | 54.3 | 54.0–54.6 | 6.95 | 53.8 | 53.3–54.3 | 7.09 |

CI = 95% confidence interval. SD = standard deviation. rVO$_{2AT}$ = oxygen uptake at anaerobic threshold relative to body mass. RER$_{AT}$ = respiratory exchange ratio at anaerobic threshold. HR$_{AT}$ = heart rate at anaerobic threshold. VE$_{AT}$ = pulmonary ventilation at anaerobic threshold. SPEED$_{AT}$ = velocity at anaerobic threshold. LA$_{AT}$ = blood lactate concentration at anaerobic threshold. rVO$_{2RCP}$ = oxygen uptake at respiratory compensation point relative to body mass. RER$_{RCP}$ = respiratory exchange ratio at respiratory compensation point. HR$_{RCP}$ = heart rate at respiratory compensation point. VE$_{RCP}$ = pulmonary ventilation at respiratory compensation point. SPEED$_{RCP}$ = velocity at respiratory compensation point. LA$_{RCP}$ = blood lactate concentration at respiratory compensation point. rVO$_{2max}$ = maximal oxygen uptake relative to body mass.

**Table 4.** Cardiopulmonary exercise testing (CPET) characteristics for cyclists.

| Variable (unit) | Derivation group n=656 | | | Testing group n=219 | | | Validation group n=219 | | |
|---|---|---|---|---|---|---|---|---|---|
| | Mean | CI | SD | Mean | CI | SD | Mean | CI | SD |
| $rVO_{2AT}$ (mL·min$^{-1}$·kg$^{-1}$) | 33.0 | 32.5–33.4 | 5.84 | 33.2 | 32.4–33.9 | 5.68 | 33.7 | 32.9–34.5 | 5.89 |
| $RER_{AT}$ | 0.87 | 0.87–0.87 | 0.04 | 0.87 | 0.87–0.88 | 0.04 | 0.87 | 0.87–0.88 | 0.04 |
| $HR_{AT}$ (beats·min$^{-1}$) | 142.2 | 141.3–143.1 | 11.7 | 140.7 | 139.1–142.3 | 11.8 | 141.2 | 139.7–142.6 | 10.8 |
| $VE_{AT}$ (L·min$^{-1}$) | 64.9 | 64.0–65.7 | 11.0 | 65.1 | 63.7–66.5 | 10.6 | 67.4 | 66.0–68.9 | 11.2 |
| $rPOW_{AT}$ (W·kg$^{-1}$) | 2.28 | 2.24–2.32 | 0.48 | 2.27 | 2.21–2.34 | 0.48 | 2.33 | 2.27–2.39 | 0.46 |
| $LA_{AT}$ (mmol·L$^{-1}$) | 1.86 | 1.82–1.90 | 0.51 | 1.84 | 1.77–1.90 | 0.50 | 1.80 | 1.74–1.87 | 0.51 |
| $rVO_{2RCP}$ (mL·min$^{-1}$·kg$^{-1}$) | 44.0 | 43.5–44.6 | 7.38 | 44.4 | 43.4–45.4 | 7.32 | 44.9 | 43.8–45.9 | 7.63 |
| $RER_{RCP}$ | 1.01 | 1.01–1.01 | 0.04 | 1.01 | 1.01–1.02 | 0.04 | 1.01 | 1.01–1.02 | 0.04 |
| $HR_{RCP}$ (beats·min$^{-1}$) | 168.8 | 168.0–169.7 | 10.5 | 167.7 | 166.2–169.2 | 11.3 | 168.4 | 167.1–169.6 | 9.11 |
| $VE_{RCP}$ (L·min$^{-1}$) | 106.2 | 104.8–107.6 | 17.7 | 107.6 | 105.3–109.8 | 16.8 | 110.4 | 107.9–112.9 | 18.7 |
| $rPOW_{RCP}$ (W·kg$^{-1}$) | 3.34 | 3.29–3.38 | 0.63 | 3.33 | 3.25–3.42 | 0.63 | 3.40 | 3.32–3.48 | 0.61 |
| $LA_{RCP}$ (mmol·L$^{-1}$) | 4.54 | 4.47–4.61 | 0.97 | 4.61 | 4.48–4.75 | 1.04 | 4.47 | 4.34–4.61 | 1.03 |
| $rVO_{2MAX}$ (mL·min$^{-1}$·kg$^{-1}$) | 51.7 | 51.1–52.4 | 7.99 | 52.0 | 50.9–53.1 | 8.01 | 52.3 | 51.2–53.4 | 8.08 |

CI = 95% confidence interval. SD = standard deviation. $rVO_{2AT}$ = oxygen uptake at anaerobic threshold relative to body mass. $RER_{AT}$ = respiratory exchange ratio at anaerobic threshold. $HR_{AT}$ = heart rate at anaerobic threshold. $VE_{AT}$ = pulmonary ventilation at anaerobic threshold. $rPOW_{AT}$ = power at anaerobic threshold relative to body mass. $LA_{AT}$ = blood lactate concentration at anaerobic threshold. $rVO_{2RCP}$ = oxygen uptake at respiratory compensation point relative to body mass. $RER_{RCP}$ = respiratory exchange ratio at respiratory compensation point. $HR_{RCP}$ = heart rate at respiratory compensation point. $VE_{RCP}$ = pulmonary ventilation at respiratory compensation point. $LA_{RCP}$ = blood lactate concentration at respiratory compensation point. $rPOW_{RCP}$ = power at respiratory compensation point relative to body mass. $rVO_{2max}$ = maximal oxygen uptake relative to body mass.

the anthropometric variables as in *Tables 1–2* and the CPET parameters given in *Tables 3–4* from a specific level of measurement (AT; RCP) and their combinations were visible.

After selection variables were included in the further analysis, only selected parameters were put into multiple linear regression (MLR) modelling. The data for MLR model building were randomly distributed into sets, that is derivation, testing, validation representing 60%, 20%, and 20% of the cases, respectively. As a result, only significant predictors (with $p<0.05$) were included in the final models. Derived equations are characterised by the coefficient of determination ($R^2$), root mean square error (RMSE), and mean absolute error (MAE). Bland-Altman plots analysis was used to establish the model's precision and accuracy during validation (*Altman and Bland, 1983*). Other implemented tests to reach the complete fulfilment of MLR modelling requirements included Ramsey's RESET test (for the correctness of specificity in MLR equations), Chow test (for stability assessment between different coefficients), and Durbin-Watson test (for autocorrelation of residuals). Each model was examined under the above-mentioned requirements and any irregularities have not been noted.

Ggplot 2 package in RStudio (R Core Team, Vienna, Austria; version 3.6.4), GraphPad Prism (GraphPad Software; San Diego, CA, USA; version 9.0.0 for Mac OS), and STATA software (StataCorp, College Station, TX, USA; version 15.1) were used in statistical analysis. A two-sided p-value <0.05 was considered as the significance borderline.

## Results
### Somatic measurements and CPET results

Anthropometric data of the runners models for derivation, testing, and validation groups are presented in *Table 1*, while cyclists are in *Table 2*. The runners groups consisted of 1998, 666, and 666 men for derivation, testing, and validation groups, respectively. In turn, the cyclists groups included 656, 219, and 219 men, respectively. Significant differences ($p<0.05$) between derivation cohorts of runners and cyclists were in BMI and age, between testing cohorts in all baseline parameters, whereas between validation cohorts only in BMI.

**Table 5.** VO$_{2max}$ prediction equations for cyclists.

| Model's category | Multiple linear regression equation | R$^2$ | Derivation group performance | | Validation group performance | |
|---|---|---|---|---|---|---|
| | | | RMSE | MAE | RMSE | MAE |
| AT | VO$_{2max}$ = 21.29 + 0.95 * rVO$_{2AT}$ + 1.74 * rPOW$_{AT}$ - 0.30 * BF | 0.811 | 3.62 | 2.89 | 3.42 | 2.72 |
| RCP | VO$_{2max}$ = 8.57 + 1.08 * rVO$_{2RCP}$ - 0.04 * VE$_{RCP}$ | 0.913 | 2.12 | 1.66 | 2.03 | 1.64 |
| AT+RCP | VO$_{2max}$ = 10.57 + 0.98 * rVO$_{2RCP}$ - 0.12 * BF | 0.909 | 2.26 | 1.78 | 2.11 | 1.72 |
| SOM | VO$_{2max}$ = 82.36–0.14 * BM - 0.66 * BF - 0.22 * Age | 0.43 | 6.06 | 4.70 | 6.11 | 4.74 |

AT = equation based on anaerobic threshold. RCP = equation based on respiratory compensation point. SOM = equation based on somatic variables only. R$^2$ = adjusted R$^2$. RMSE = root mean square error. MAE = mean absolute error (mL·min$^{-1}$·kg$^{-1}$). VO$_{2max}$ = maximal oxygen uptake relative to body mass (mL·min$^{-1}$·kg$^{-1}$). rVO$_{2AT}$ = oxygen uptake at anaerobic threshold relative to body mass (mL·min$^{-1}$·kg$^{-1}$). rPOW$_{AT}$ = power at anaerobic threshold relative to body mass (W·kg$^{-1}$). rVO$_{2RCP}$ = oxygen uptake at respiratory compensation point relative to body mass (mL·min$^{-1}$·kg$^{-1}$). VE$_{RCP}$ = pulmonary ventilation at respiratory compensation point (L·min$^{-1}$). BF = body fat (%). BM = body mass (kg).

CPET results for runners models are presented in *Table 3* and for cyclists in *Table 4*. Runners in the derivation cohort achieved relative to body mass VO$_{2max}$ (rVO$_{2max}$) of 53.8±6.67 mL·min$^{-1}$·kg$^{-1}$ (95% CI: 53.3–54.3), in testing group 54.3±6.95 mL·min$^{-1}$·kg$^{-1}$ (95% CI: 54.0–54.6), and in validation group 53.8±7.09 mL·min$^{-1}$·kg$^{-1}$ (95% CI: 53.3–54.3). In cyclists groups mean rVO$_{2max}$ was 51.7±7.99 mL·min$^{-1}$·kg$^{-1}$ (95% CI: 51.1–52.4), 52.0±8.01 mL·min$^{-1}$·kg$^{-1}$ (95% CI: 50.9–53.1), and 52.3±8.08 mL·min$^{-1}$·kg$^{-1}$ (95% CI: 51.2–53.4) for derivation, testing, and validation cohorts, respectively. Relative to body mass oxygen uptake at anaerobic threshold (rVO$_{2AT}$) in runners groups accounted for 71.6 ± 5.10% (95% CI: 71.2–71.9), 71.1 ± 4.91% (95% CI: 70.9–71.3), and 71.0 ± 5.42% (95% CI: 70.6–71.5) of rVO$_{2max}$ in derivation, testing, and validation cohorts, respectively. In cyclists, it was 63.7 ± 5.20% (95% CI: 63.3–64.1%), 63.8 ± 5.00% (95% CI: 63.1–64.4), and 64.4 ± 4.73% (95% CI: 63.8–65.0) of rVO$_{2max}$, respectively. In turn, relative to body mass oxygen uptake at respiratory compensation point (rVO$_{2RCP}$) in runners accounted for 87.8 ± 3.23% (95% CI: 87.5–88.0), 87.6 ± 3.60% (95% CI: 87.4–87.8), and 87.7 ± 3.35% (87.4–88.0) of rVO$_{2max}$ for derivation, testing, and validation cohorts, respectively, while in cyclists for 85.0 ± 4.18% (95% CI: 84.7–85.4), 85.4 ± 4.09% (95% CI: 84.8–85.9), and 85.7 ± 4.14% (95% CI: 85.1–86.2) of rVO$_{2max}$, respectively. There were no significant differences in threshold-to-maximum percentages values between derivation, testing, and validation cohorts among the runners and cyclists groups, whereas variations between runners and cyclists threshold-to-maximum results were all significant (p<0.05).

## Prediction models based on AT and RCP

Full forms of MLR prediction models for cyclists are demonstrated in *Table 5*, whereas for runners in *Table 6*. The models prediction performance is presented as R$^2$ along with RMSE and MAE. Briefly,

**Table 6.** VO$_{2max}$ prediction equations for runners.

| Model's category | Multiple linear regression equation | R$^2$ | Derivation group performance | | Validation group performance | |
|---|---|---|---|---|---|---|
| | | | RMSE | MAE | RMSE | MAE |
| AT | VO$_{2max}$ = 19.78 + 1.05 * rVO$_{2AT}$ + 0.94 * SPEED$_{AT}$ - 0.12 * FFM - 0.06 * VE$_{AT}$ - 0.07 * HR$_{AT}$ | 0.775 | 3.43 | 2.61 | 3.60 | 2.74 |
| RCP | VO$_{2max}$ = 1.98 + 1.03 * rVO$_{2RCP}$ + 0.23 * SPEED$_{RCP}$ | 0.899 | 2.0 | 1.58 | 2.08 | 1.60 |
| AT+RCP | VO$_{2max}$ = 1.98 + 1.03 * rVO$_{2RCP}$ + 0.23 * SPEED$_{RCP}$ | 0.899 | 2.0 | 1.58 | 2.08 | 1.60 |
| SOM | VO$_{2max}$ = 72.37–0.77 * BF - 0.19 * Age | 0.35 | 5.53 | 4.36 | 5.54 | 4.37 |

AT = equation based on anaerobic threshold. RCP = equation based on respiratory compensation point. SOM = equation based on somatic variables only. R$^2$ = adjusted R$^2$. RMSE = root mean square error. MAE = mean absolute error (mL·min$^{-1}$·kg$^{-1}$). VO$_{2max}$ = maximal oxygen uptake relative to body mass (mL·min$^{-1}$·kg$^{-1}$). rVO$_{2AT}$ = oxygen uptake at anaerobic threshold relative to body mass (mL·min$^{-1}$·kg$^{-1}$). SPEED$_{AT}$ = velocity at anaerobic threshold (km·h$^{-1}$). FFM = fat free mass (kg). VE$_{AT}$ = pulmonary ventilation at anaerobic threshold (L·min$^{-1}$). HR$_{AT}$ = heart rate at anaerobic threshold (beats·min$^{-1}$). BF = body fat (%). rVO$_{2RCP}$ = oxygen uptake at respiratory compensation point relative to body mass (mL·min$^{-1}$·kg$^{-1}$). SPEED$_{RCP}$ = velocity at respiratory compensation point (km·h$^{-1}$).

$R^2$ ranged for cyclists equations from 0.43 for somatic parameters (SOM) equation to 0.913 for RCP equations. For runners formulae, $R^2$ ranged from 0.35 for SOM equation to 0.899 for AT and AT+RCP equations. Obtained RMSE for cyclists models was the lowest for RCP equations (=2.03) and the highest for SOM equation (=6.11). For runners, RMSE ranged from 2.0 for AT and AT+RCP equations to 5.54 for SOM equation. Similarly, observed MAE for cyclists was the lowest for RCP equation (=1.64 mL·min$^{-1}$·kg$^{-1}$) in the validation group and the highest for SOM equation (=4.74 mL·min$^{-1}$·kg$^{-1}$), while in runners the lowest for AT and AT+RCP equations (=1.58 mL·min$^{-1}$·kg$^{-1}$) and the highest for SOM equation (=4.37 mL·min$^{-1}$·kg$^{-1}$). The performance of prediction equations is demonstrated in *Figure 2*.

### Models validation

Evaluation of each model for cyclists is presented in *Table 5*, while for runners in *Table 6*. In summary, the performance of our prediction equations was similar to that observed in the derivation cohort. A minorly higher RMSE and MAE were noted. Overall, RMSE values in cyclists are located between 2.03 and 6.11, whereas in runners between 2.0 and 5.54. MAE ranged from 1.64 to 4.74 mL·min$^{-1}$·kg$^{-1}$ in cyclists models and 1.58 to 4.37 mL·min$^{-1}$·kg$^{-1}$ in runners. The most accurate prediction was obtained in cyclists (defined as the highest replicability and the lowest risk of inaccuracies in the test set) by RCP equations ($R^2$=0.913, RMSE=2.03, MAE=1.64). Interestingly, the models which worked the most accurately and the less precisely were the same in the derivation and validation. *Figure 3* illustrates the Bland-Altman plots with a comparison of observed vs predicted VO$_{2max}$ using newly derived prediction models at the stage of validation.

## Discussion

In the present study, we derived and internally validated novel advanced and accurate prediction models for VO$_{2max}$. The main findings are as follows: (1) we can precisely predict VO$_{2max}$ based on submaximal CPET parameters, (2) inclusion of cardiopulmonary and BC variables enriches their prediction performance, (3) based only on somatic parameters, weak-to-low VO$_{2max}$ assessment is currently possible, (4) derived equations showed high transferability abilities during validation. Our findings indicate that prediction models based on AT and RCP variables allow for accurate VO$_{2max}$ calculation. Equations based on somatic variables allow for limited precision.

The main advantage of our research is the unified CPET protocol conducted on a wide cohort of endurance athletes with different levels of fitness. This approach enables the comprehensive evaluation of the most important predictors which were further applied to build prediction equations. In the current literature, prediction models for sports and performance diagnostics are mostly derived from narrow and specified athletic cohorts which limit their applicability to broader populations (*Paap and Takken, 2014*). Moreover, the advantage of the presented research was the fact that regression equations for the treadmill and cycle ergometer were derived based on the most commonly used machines and forms of activity or movement in the laboratory stress exercise tests.

An important issue addressed in publications on various attempts to estimate VO$_{2max}$ is the question of their usefulness in assessing changes in endurance over a training cycle (*Klusiewicz et al., 2016*). As reported by Klusiewicz et al., the suitability of the two indirect methods of assessing VO$_{2max}$ was statistically confirmed, their usefulness for estimating changes in the endurance of the trained individuals during the training cycle was rather low (*Klusiewicz et al., 2016*). The standard estimation error of these methods (ranged between 4.2% and 7.7% in the female and 5.1% and 7.4% in the male) was higher than the real differences in the VO$_{2max}$ values determined in the direct measurements (between the first and the second examination the VO$_{2max}$ rose by 3.0% in the female athletes and dropped by 4.3% in the male athletes) (*Klusiewicz et al., 2016*). Popularly used wearables provided substantial accuracy on population level when considered devices with exercise-based algorithms (*Molina-Garcia et al., 2022*). However, VO$_{2max}$ predictions on the individual's level still need improvement in the context of both sports and clinical settings (*Molina-Garcia et al., 2022*).

In the Astrand-Ryhming method, a widely used VO$_{2max}$ prediction method for almost 70 years, in several papers published so far, the correlation coefficients of the measured values to the predicted values ranged from 0.63 to 0.85. Standard estimated error values (in L·min$^{-1}$) generally exceeded 0.5 (*Grant et al., 1995*; *Legge and Banister, 1986*). In our study, the highest $R^2$ was 0.913 and the

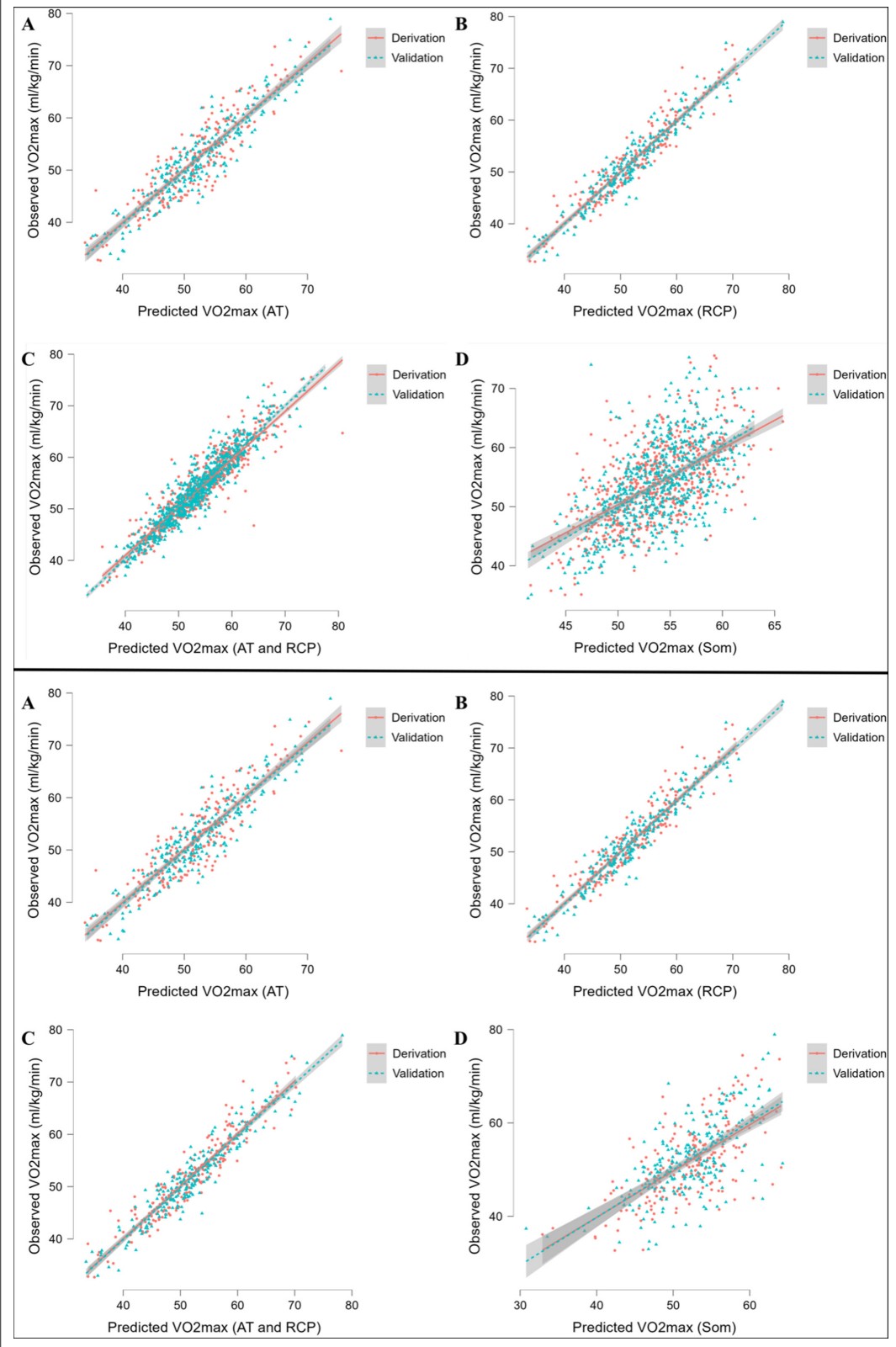

**Figure 2.** Performance of prediction equations for VO$_{2max}$. Abbreviations: VO$_{2max}$; maximal oxygen uptake; AT, anaerobic threshold; RCP, respiratory compensation point; Max, maximal; Som, somatic. All values are presented in mL·min$^{-1}$·kg$^{-1}$. Upper panel shows performance for running equations, while the lower panel shows performance for cycling equations. Panel A shows performance of the prediction model for AT; panel B for RCP; panel C for AT and RCP; panel D for somatic-only equation.

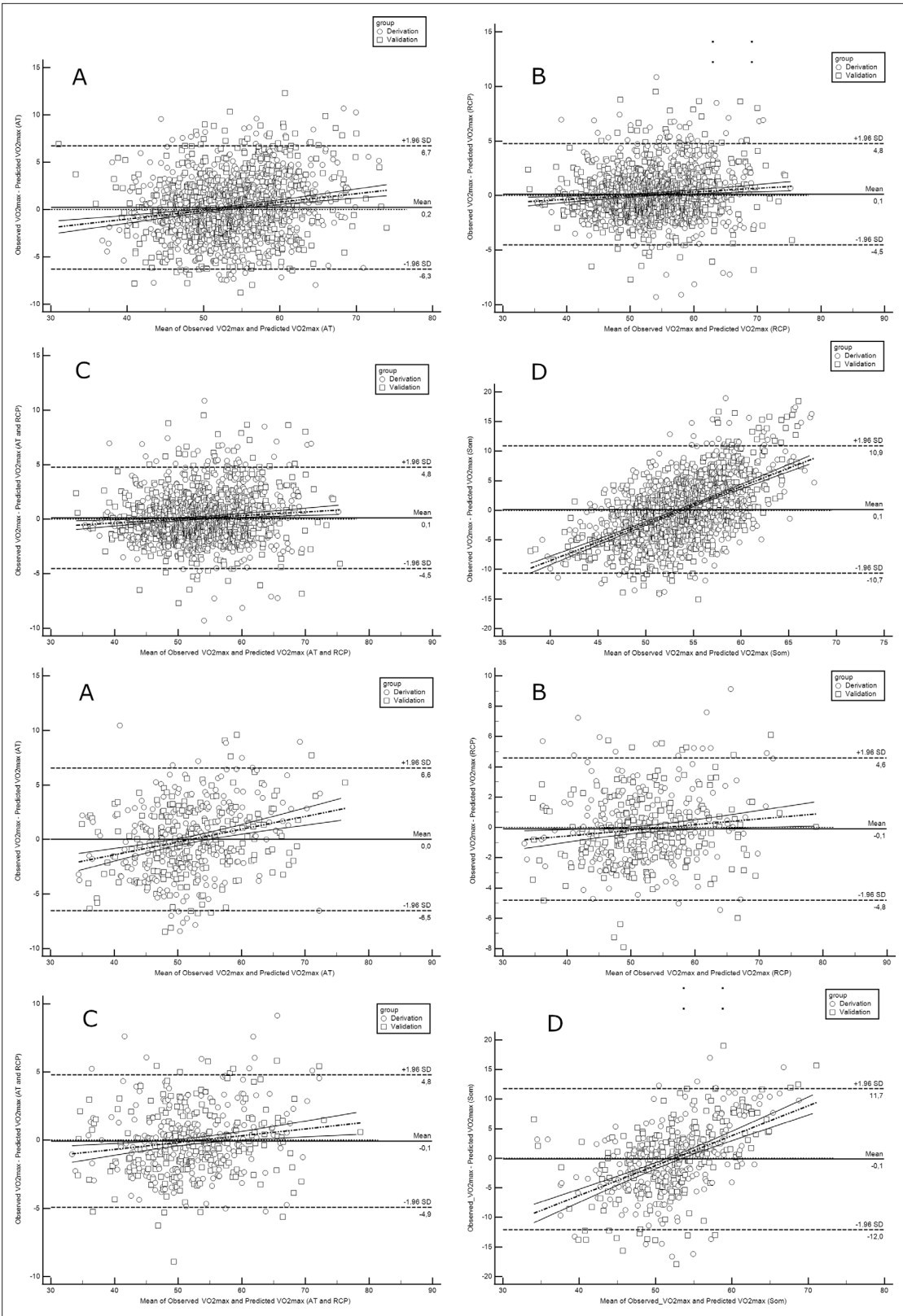

**Figure 3.** Bland-Altman plots comparing observed with predicted VO$_{2max}$ in runners derivation and validation cohorts. Abbreviations: VO$_{2max}$; maximal oxygen uptake; AT, anaerobic threshold; RCP, respiratory compensation point; Max, maximal; Som, somatic. All values are presented in mL·min$^{-1}$·kg$^{-1}$. Upper panel shows performance for running equations, while the lower panel shows performance for cycling equations. Panel A shows performance of the prediction model for AT; panel B for RCP; panel C for AT and RCP; panel D for the somatic-only equation.

lowest was 0.775. As an additional advantage, we propose an equation based only on somatic variables, which showed a low $R^2$ - 0.35 for runners and 0.43 for cyclists. Although, it still presents that our models are more accurate than those widely described in the literature so far (*Paap and Takken, 2014*; *Wiecha et al., 2023*).

As $VO_2$ is an exercise parameter that combines the function of the respiratory, circulatory, and muscular systems, the use of only somatic variables such as body fat, weight, or age was not sufficient for optimal prediction (*Bassett and Howley, 2000*). It is worth underlining that the main factors contributing to $VO_{2max}$ were submaximal variables $VO_{2AT}$ and $VO_{2RCP}$, as well as running speed for runners and pedalling power for cyclists (*Billat and Koralsztein, 1996*). Thus, $VO_2$ is a universal and interchangeable measurement of endurance (*Albouaini et al., 2007*). Our results suggest that $VO_2$ is an indicator of both endurance and critical vital signs (*Blair et al., 1989*; *Kaminsky et al., 2015*). It is also in line with current standings as Kaminsky et al. and Blair et al. postulate that the higher the $VO_{2max}$, the fitter the individual is (*Kaminsky et al., 2015*), and the lower its all-cause mortality (*Blair et al., 1989*).

In sports medicine and exercise physiology, evaluation of the body's functional performance remains crucial (*Sartor et al., 2013*). Our results indicate that $VO_{2max}$ was possible to accurately predict based only on submaximal parameters (without the inclusion of maximal ones), which is reflected in $R^2$ (*Wiecha et al., 2022*). This confirms that submaximal CPET is a valuable tool in assessing fitness levels. Thus, submaximal exercise testing appears to be more applicable by physicians and fitness professionals in their role as clinical exercise specialists (*Noonan and Dean, 2000*). For individuals who have a moderate to high possibility of cardiovascular diseases, exerting themselves up to their maximum abilities increases risk of adverse outcomes (*Guazzi et al., 2012*; *Noonan and Dean, 2000*). There are numerous possibilities to use results of submaximal CPET. Currently, pseudo-random binary sequencing appears as one of the feasible approaches. Its enables assessment of cardiorespiratory kinetics within the selected workload ranges (*Hoffmann et al., 2022*; *Koschate et al., 2016*). This is important as CPET until refusal is often impossible to conduct or highly dangerous (*Guazzi et al., 2016*). Such situations appear mainly in clinical cardiovascular conditions, such as heart failure, dyspnea of unknown aetiology, or risk evaluation for providing treatment protocol (*Guazzi et al., 2016*). Furthermore, previous studies have described that submaximal variables are significant predictors for performance measurements, as pointed out by *Snowden et al., 2010*, and *Albouaini et al., 2007*. In conclusion, ensuring high repeatability through submaximal prediction methods is crucial for monitoring endurance changes in both sports and medical diagnostics (*Mann et al., 2013*; *Noonan and Dean, 2000*).

It is worth mentioning the effect of body fat percentage on $VO_{2max}$. This variable has been included in the majority of our models. With the increase in body fat percentage, $VO_2$ decreased, and this relationship was particularly important in the somatic equation and is previously described in the literature (*Shete et al., 2014*). This is due to the fact that a higher level of adipose tissue and general body mass have both a negative impact on the results during long-term endurance sports (i.e. running and cycling) and with increasing fitness levels, the level of participant fatness decreases (*Schwartz et al., 1991*).

Results of internal validation show that our prediction models allow for an accurate assessment of $VO_{2max}$. The observed RMSE and MAE values are significantly lower than in the validation of other prediction models on endurance athletes' cohorts. *Petek et al., 2022*, and *Malek et al., 2004*, while validating the majority of widely used prediction models, observed MAE and RMSE on the level of 7–9 mL·kg$^{-1}$·min$^{-1}$. Our highest value of error for the somatic equation was in the cycling model (MAE; 4.74 mL·kg$^{-1}$·min$^{-1}$). Moreover, as we mentioned above, the somatic equation showed the lowest accuracy, and the remaining equations have RMSE between 1.94 and 6.11 and MAE in the range of 1.46–4.74 mL·kg$^{-1}$·min$^{-1}$.

Our study has some limitations. The applied exercise protocol may affect CPET results. There may be differences in performance measured in 2 min steps comparing to longer steps, but this should not significantly impact the participants' exercise results. Additionally, longer constant intervals may increase accuracy in determining AT and RCP level, but have negligible impact on $VO_{2max}$ values (*Muscat et al., 2015*). The study, due to the insufficient number of women in the database to obtain reliable results, was restricted to men only. Therefore, the equations should be applied with more caution in women.

To summarise, our study has vast practical applications in the comprehensive assessment of an athlete's training and is a valuable tool for coaches in the preparation of individualised training prescriptions (*Mann et al., 2013*). Targeting training regimens and diet to optimise the most important parameters contributing to the $VO_{2max}$ (i.e. $VO_{2AT}$, $VO_{2RCP}$, RER, body fat, etc.) will allow for achieving better results during the competition and they provide a useful indirect method for assessing changes in endurance during the training cycle (*Mann et al., 2013*). Various areas of application of prediction models have also been postulated in the literature so far, for example in submaximal and maximal efforts, or simulating the overcoming of the starting distance, or even at rest (*Zhou et al., 1997*). It is also worth mentioning their clinical implications in cardiology for the diagnosis of heart disease in athletes (where a reduction in $VO_{2max}$ may occur despite maintaining other parameters, e.g. RER) (*Guazzi et al., 2016*; *Löllgen and Leyk, 2018*).

## Conclusion

Briefly, we provided new prediction models for $VO_{2max}$. The proposed method allows for precise prediction of $VO_{2max}$ based on submaximal results. Our equations were derived from a wide cohort of 6439 athletes with varied fitness levels which inflated the quality and transferability of the presented data. Higher accuracy was noted when applying submaximal predictors. Adding circulatory and respiratory variables enriches prediction performance. Body fat and fat-free mass had significant impacts on most of the $VO_{2max}$ prediction equations. The novel model based only on somatic parameters is presented. Derived equations showed high performance during internal validation and were fairly replicable. The inclusion of such a tool has practical usage for fitness professionals and personal coaches to prepare more precise training recommendations and establish competition pacing strategies.

---

## Additional information

### Competing interests

Szczepan Wiecha: received payment for leading CPET workshops at IX Małopolskich Warsztatach Niewydolności Serca. The author has no other competing interest to declare. Tomasz Kowalski: has received funding from the Institute of Sport - National Research Institute. The author has received consulting fees for regular coaching and consulting work with private clients, Polish Triathlon Federation and The Triathlon Squad professional triathlon team. The author has no other competing interests to declare. The other authors declare that no competing interests exist.

### Funding

No external funding was received for this work.

### Author contributions

Szczepan Wiecha, Conceptualization, Resources, Data curation, Formal analysis, Validation, Investigation, Visualization, Methodology, Writing – original draft, Project administration, Writing – review and editing; Przemysław Seweryn Kasiak, Piotr Szwed, Data curation, Writing – original draft; Tomasz Kowalski, Data curation, Methodology, Writing – original draft; Igor Cieśliński, Data curation, Software, Formal analysis, Validation; Marek Postuła, Andrzej Klusiewicz, Supervision, Writing – review and editing

### Author ORCIDs

Szczepan Wiecha ⓘ http://orcid.org/0000-0001-9458-557X
Przemysław Seweryn Kasiak ⓘ http://orcid.org/0000-0002-0303-6135

### Ethics

Human subjects: The Institutional Review Board of the Bioethical Committee at the Medical University of Warsaw (AKBE/32/2021) has approved the study protocol. The regulations of the Declaration of Helsinki were met during all parts of the study. Each subject delivered written consent to undergo CPET and participate in the study.

Decision letter and Author response
Decision letter https://doi.org/10.7554/eLife.86291.sa1
Author response https://doi.org/10.7554/eLife.86291.sa2

## Additional files

### Supplementary files
- MDAR checklist
- Source code 1. Source code in Python for transforming files in the database.
- Reporting standard 1. TRIPOD checklist.

### Data availability
All data generated or analysed during this study are included in the manuscript.

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
