## [Editor Report]

The authors have established new formulas to predict maximum oxygen uptake for cyclists and runners based on submaximal exercise testing and anthropometric characteristics. This is an important study with a large and comprehensive dataset, which may be helpful for many exercise labs. The work is convincing, using appropriate and validated methodology in line with the current state-of-the-art, as shown by references to common exercise books.

---

## [Decision Letter]

**Decision letter after peer review:**

Thank you for submitting your article "VO2max prediction based on submaximal cardiorespiratory relationships and body composition in male runners and cyclists: a population study" for consideration by *eLife*. Your article has been reviewed by 3 peer reviewers, including Herbert Löllgen as the Reviewing Editor and Reviewer #1, and the evaluation has been overseen by Matthias Barton as the Senior Editor. The following individual involved in the review of your submission has agreed to reveal their identity: Beat Knechtle (Reviewer #3).

The reviewers have discussed their reviews with one another, and the Reviewing Editor has drafted this to help you prepare a revised submission. Please follow all recommendations.

*Reviewer #1 (Recommendations for the authors):*

After going through the manuscript and reviewing the work, some comments arise:

Figure 3: The legends are not readable and should be improved.

A comment on the wide scatter is necessary, the correlations here are not clinically significant because of the large variation.

Table 6 the abbreviations RCP and SOM should be listed for better understanding.

The aim of the work: determination of the maximum performance from submaximal values should be better presented in the discussion as this is of great importance for the reader.

The work can then be accepted after changes.

There should be a short final conclusion for the reader like a take-home message and what this study adds and will support daily practice.

*Reviewer #2 (Recommendations for the authors):*

Apart from my objection concerning the missing limitations I like this work.

Nevertheless, the manuscript should be carefully revised.

For example: Lines 58-76 address the problem that it is not always possible to undertake a test to exhaustion (athletes, patients). The authors failed to point to PRBS-testing (pseudo-random binary sequence) allowing the determination of VO2 kinetics during very mild exercise intensities (e.g. 20W and 80 W).

*Reviewer #3 (Recommendations for the authors):*

Line 17: add in the abstract what is not known to justify the study.

Line 19: Explain the abbreviation CPET.

Lines 53-55: add a reference.

Lines 59-60: add a reference.

Line 66: add a reference.

Line 71: add a reference.

Line 77: add a reference.

Lines 80-81: add a reference.

Lines 81-82: add a reference.

Line 94: what is the hypothesis of your study?

Line 131: what is 'proper preparation'?

Line 266: can you confirm your hypothesis?

Lines 276-277: add a reference.

Lines 284-286: add a reference.

Line 340: strength of the study? Limitations of the study?

---

## [Author Response]

Reviewer #1 (Recommendations for the authors):After going through the manuscript and reviewing the work, some comments arise:Figure 3: The legends are not readable and should be improved.

Figure 3 shows a panel of Blant-Altman plots for which no numerical/statistical reporting is given, this is only a visual assessment to ascertain agreement. The higher-resolution figures will be adjusted during the further processing of the manuscript. The separate files in the system are high resolution and allow a better reading of the data after magnification which cannot be achieved at the current review stage.

A comment on the wide scatter is necessary, the correlations here are not clinically significant because of the large variation.Table 6 the abbreviations RCP and SOM should be listed for better understanding.

We have added explanations of abbreviations.

The aim of the work: determination of the maximum performance from submaximal values should be better presented in the discussion as this is of great importance for the reader.

We have supplemented the discussion to highlight the validity and predictability of VO2max based on submaximal CPET values.

The work can then be accepted after changes.There should be a short final conclusion for the reader like a take-home message and what this study adds and will support daily practice.

We added a summary of the most important results from our work in the editorial system as an "Impact Statement".

Reviewer #2 (Recommendations for the authors):Apart from my objection concerning the missing limitations I like this work.Nevertheless, the manuscript should be carefully revised.For example: Lines 58-76 address the problem that it is not always possible to undertake a test to exhaustion (athletes, patients). The authors failed to point to PRBS-testing (pseudo-random binary sequence) allowing the determination of VO2 kinetics during very mild exercise intensities (e.g. 20W and 80 W).

Thank you for drawing our attention to a very interesting study on the possibility of predicting VO2 and HR from analysis of cardiorespiratory kinetics in random low-intensity exercise. We have added information about this possibility in the manuscript because of its potential usefulness by using very low loads. We think that the method could be particularly useful in clinical diagnosis once appropriately validated.

Reviewer #3 (Recommendations for the authors):Line 17: add in the abstract what is not known to justify the study.

Thank you for pointing this out, we have added the information in the abstract.

Line 19: Explain the abbreviation CPET.

We have expanded the abbreviation.

Lines 53-55: add a reference.Lines 59-60: add a reference.Line 66: add a reference.Line 71: add a reference.Line 77: add a reference.Lines 80-81: add a reference.Lines 81-82: add a reference.

We have added a reference to the above sections.

Line 94: what is the hypothesis of your study?

As suggested, we have added a hypothesis at the end of the paragraph.

Line 131: what is 'proper preparation'?

We have added more detailed information on our standard CPET preparation procedure.

Line 266: can you confirm your hypothesis?

We have added a relevant statement.

Lines 276-277: add a reference.Lines 284-286: add a reference.

We have added a reference to the above sections.

Line 340: strength of the study? Limitations of the study?

We have added information in the final part of the discussion before the summary.